# Brain interstitial pH changes in the subacute phase of hypoxic-ischemic encephalopathy in newborn pigs

Gábor Remzső[1]☯*, János Németh[1]☯, Viktória Varga[1], Viktória Kovács[1], Valéria Tóth-Szűki[1], Kai Kaila[2,3], Juha Voipio[2], Ferenc Domoki[1]

**1** Department of Physiology, University of Szeged, Faculty of Medicine, Szeged, Hungary, **2** Faculty of Biological and Environmental Sciences, Molecular and Integrative Biosciences, University of Helsinki, Helsinki, Finland, **3** Neuroscience Center (HiLIFE), University of Helsinki, Helsinki, Finland

☯ These authors contributed equally to this work.
\* remzso.gabor@med.u-szeged.hu

**Data Availability Statement:** All experimental data are available at Open Science Framework (osf.io): DOI 10.17605/OSF.IO/MUTGA.

## Abstract

Brain interstitial pH ($pH_{brain}$) alterations play an important role in the mechanisms of neuronal injury in neonatal hypoxic-ischemic encephalopathy (HIE) induced by perinatal asphyxia. The newborn pig is an established large animal model to study HIE, however, only limited information on $pH_{brain}$ alterations is available in this species and it is restricted to experimental perinatal asphyxia (PA) and the immediate reventilation. Therefore, we sought to determine $pH_{brain}$ over the first 24h of HIE development in piglets. Anaesthetized, ventilated newborn pigs (n = 16) were instrumented to control major physiological parameters. $pH_{brain}$ was determined in the parietal cortex using a pH-selective microelectrode. PA was induced by ventilation with a gas mixture containing $6\%O_2$-$20\%CO_2$ for 20 min, followed by reventilation with air for 24h, then the brains were processed for histopathology assessment. The core temperature was maintained unchanged during PA (38.4±0.1 *vs* 38.3±0.1˚C, at baseline *versus* the end of PA, respectively; mean±SEM). In the arterial blood, PA resulted in severe hypoxia ($P_aO_2$: 65±4 *vs* 23±1\*mmHg, \*p<0.05) as well as acidosis ($pH_a$: 7.53 ±0.03 *vs* 6.79±0.02\*) that is consistent with the observed hypercapnia ($P_aCO_2$: 37±3 *vs* 160 ±6\*mmHg) and lactacidemia (1.6±0.3 *vs* 10.3±0.7\*mmol/L). Meanwhile, $pH_{brain}$ decreased progressively from 7.21±0.03 to 5.94±0.11\*. Reventilation restored $pH_a$, blood gases and metabolites within 4 hours except for $P_aCO_2$ that remained slightly elevated. $pH_{brain}$ returned to 7.0 in 29.4±5.5 min and then recovered to its baseline level without showing secondary alterations during the 24 h observation period. Neuropathological assessment also confirmed neuronal injury. In conclusion, in spite of the severe acidosis and alterations in blood gases during experimental PA, $pH_{brain}$ recovered rapidly and notably, there was no post-asphyxia hypocapnia that is commonly observed in many HIE babies. Thus, the neuronal injury in our piglet model is not associated with abnormal $pH_{brain}$ or low $P_aCO_2$ over the first 24 h after PA.

**Funding:** JN,GR,VTSZ,VV,VK,FD; 2.0 1.3 2017 1,2.1 NKP 2017 00002; Hungarian Brain Research Program JN,GR,VTSZ,VV,VK,FD; EFOP-3.6.1-16-2016-00014; EU-funded Hungarian grant EFOP JN, GR,VTSZ,VV,VK,FD; GINOP 2.3.2. 15 2016 00034; GINOP The funders had no role in study design, data collection and analysis, decision to publish, or preparation of the manuscript.

**Competing interests:** The authors have declared that no competing interests exist.

## Introduction

Perinatal asphyxia (PA), defined as $O_2$ deprivation around the time of delivery, is one of the primary causes of neonatal morbidity and mortality worldwide, affecting ~4 million neonates annually [1]. The interruption of the placental or pulmonary gas exchange induces immediate metabolic changes (hypoxemia, hypercapnia and mixed acidosis) that trigger cardiovascular responses in favour of maintaining $O_2$ delivery to the myocardium and the brain (i.e. centralized circulation). When these compensatory mechanisms are exhausted, critical tissue hypoxia/ischemia will occur, and the subsequent metabolic crisis will lead to hypoxic-ischemic encephalopathy (HIE) in the survivors [2].

Despite the partial efficacy of therapeutic hypothermia to mitigate the adverse outcome of PA/HIE [3,4], preclinical animal models are still required to study the precise pathophysiology of HIE development, and also to test putative neuroprotective approaches [5]. Similar to the heterogeneity of the aetiology, severity, and duration of the human PA/HIE syndrome, there is also considerable heterogeneity in the preclinical models, both in the species, the prenatal/postnatal age of the animals, the methodology to induce PA/HIE, and the length of the post-asphyxial observation period. These differences do not allow easy comparison and translation of the accumulated data. The newborn pig has long been identified and accepted as a general preclinical model to study the term human neonate [6,7]. There are a number of important similarities between the two species in brain structure, metabolism, development [8,9], and in cerebrovascular physiology [10] that make piglets adequate for translational PA/HIE research. We have recently published a newborn piglet PA/HIE model [11] that produced major hallmarks of human PA/HIE and was able to elicit significant neuronal injury, but did not include manipulations that occur never or very rarely in humans such as bilateral carotid artery occlusion or transient hemorrhagic hypotension.

In the present study, we set out to investigate the course of brain interstitial pH ($pH_{brain}$) changes during PA and the subacute phase of HIE development in our piglet model. In HIE patients using magnetic resonance spectroscopy (MRS) techniques at < 2 weeks of age, brain intracellular pH ($pH_i$) levels were found to be rather elevated [12]. This alkalosis persisted for months and correlated with adverse neurological outcome, and the described pH alteration may be one of the features of the so-called secondary energy failure [13,14]. However, changes in $pH_i$ may not truthfully reflect $pH_{brain}$ alterations as transmembrane pH gradients are subject to change during HIE development, and furthermore, intracellular and extracellular pH have different targets via which they may change brain activity and energy consumption [15]. There are very limited data on $pH_{brain}$ changes elicited by PA/HIE from piglets [16], and these studies did not follow the course of $pH_{brain}$ beyond 4 hours after the completion of the hypoxic-ischemic stress. We now report quantitative cerebrocortical $pH_{brain}$ data in our well-characterized piglet PA/HIE model both during the PA and the first 24 hours of HIE development.

## Materials and methods

All experimental procedures involving animals were approved in a three-step process. First, the detailed experimental plan was carefully reviewed and approved by the Institutional Animal Care and Use Committee of the University of Szeged (IACUC, in Hungarian: SZTE Munkahelyi Állatjóléti Bizottság). Second, the approval of the IACUC-endorsed experimental plan was requested from the National Ethical Committee on Animal Experiments (in Hungarian: Állatkísérletes Tudományos Etikai Tanács, ÁTET). Third, the National Food Chain Safety and Animal Health Directorate of Csongrád county, Hungary on behalf of the Hungarian Government issued the permit based on the ÁTET recommendation (permit nr: XIV./1414/2015). All animal experiments complied with (1) the guidelines of the Scientific Committee of Animal

Experimentation of the Hungarian Academy of Sciences (updated Law and Regulations on Animal Protection: 40/2013. (II. 14.) Gov. of Hungary), (2) the EU Directive 2010/ 63/EU on animal protection used for scientific research, and (3) with the ARRIVE guidelines. Furthermore, we state that all procedures in the present study were performed on anesthetized animals, also that all animals were anesthetized and treated with analgetics along with intensive monitoring of vital signs throughout the observation period (24 h) without expected or observed mortality. At the end of the experiments the animals were euthanized with an overdose of pentobarbital sodium (300 mg, Release; Wirtschaftsgenossenschaft deutscher Tierärzte eG, Garbsen, Germany).

Newborn (≤1 day old) male Landrace pigs (body weight: 1.5–2.5 kg, n = 16) were obtained from a local company (Pigmark Ltd., Co., H-6728, Rózsamajor út 13., Szeged, Hungary) and delivered to the laboratory on the morning of the experiments. Anaesthesia was induced by sodium thiopental (45 mg/kg ip; Sandoz, Kundl, Austria). Piglets were intubated via tracheotomy and artificially ventilated by a pressure-controlled small animal respirator with warmed, humidified medical air (21% $O_2$, balance $N_2$) that could be optionally supplemented with $O_2$. Respiratory settings (fraction of inspired $O_2$: 0.21–0.25; respiratory rate (RR): 30–35 1/min, peak inspiratory pressure: 120–135 $mmH_2O$) were adjusted to maintain blood gas values and $O_2$ saturation in the physiologic range. The right femoral vein was catheterized under aseptic conditions to maintain anaesthesia/analgesia with a bolus injection of morphine (100 μg/kg; Teva, Petach Tikva, Israel) and midazolam (250 μg/kg; Torrex Pharma, Vienna, Austria), then with continuous infusion of morphine (10 μg/kg/h), midazolam (250 μg/kg/h) and fluids (5% glucose, 0.45% NaCl 3–5 ml/kg/h). A second catheter was placed into the right carotid artery for continuous monitoring of mean arterial blood pressure (MABP) and heart rate (HR). This artery was chosen as ligation of the femoral artery would have resulted in critical ischemia of the hindlimb over the 24h reventilation period (personal observations), in contrast, unilateral carotid artery occlusion has been shown not to affect cerebral blood flow [17]. Rectal temperature was measured continuously and kept in the physiologic range (38.5±0.5˚C) with a servo-controlled water circulation heating-cooling pads (Blanketrol III., Cincinnati Sub-Zero, Cincinnati, Ohio, USA). $O_2$ saturation, MABP and HR were continuously monitored using a Hewlett-Packard M1094 monitor (Palo Alto, California, USA) and recorded online (Mecif-View, Arlington, Mass., USA). These parameters were recorded at baseline, during PA and and then for 10 minutes at the beginning of each reventilation hour. Arterial blood samples (~300 μl/sample) were analysed with a blood analysis system (EPOC Blood Analysis, Epocal Inc., Ottawa Canada) at baseline, at the end of asphyxia; then at selected intervals up to 20 hours to determine arterial blood pH, gas tensions, base excess, central oxygen saturation, hemoglobin, bicarbonate, glucose and lactate concentrations. Prophylactic antibiotics were given iv.: penicillin (50 mg/kg/12 h, Teva, Petah Tikva, Israel) and gentamicin (2.5 mg/kg/12 h, Sanofi, Paris, France). The urinary bladder was tapped by suprapubic puncture at 12 hour after asphyxia.

## $pH_{brain}$ measurements

Measurements were performed inside a self-made Faraday cage with 4 Hz sampling rate. pH-selective microelectrodes (external tip diameter: 50 μm) were obtained from Unisense (Aarhus, Denmark), whereas glass reference microelectrodes (external tip diameter: ~20 μm and filled with 150 mM NaCl; resistance: ~4-5x10^10 Ohm) were self-made and used with Ag/AgCl wire electrodes. The electrodes were mounted on stereotaxic manipulators for calibration in 3 different warmed (38˚C) buffer solutions (pH: 6.10, 7.10, and 8.10, respectively) before each experiment. The piglet head was fixed in a stereotaxic frame and after retracting the scalp, two

small circular craniotomies ($\varnothing \cong 5$ mm) were made over the fronto-parietal cortex, and the dura mater was gently removed. The tips of the pH and reference microelectrodes were installed ~1–2 mm deep into the exposed cortex, and a Ag/AgCl ground electrode was placed under the scalp. The electrode signals were recorded, digitized and stored either using a custom-built differential electrometer ($>10^{14}$ Ohm input impedance; 16 Hz low pass cut-off), a 16-bit analog-to-digital converter (National Instruments, Austin, TX) and WinEDR software (Dr. John Dempster, University of Strathclyde, UK), or using a Microsensor Multimeter and SensorTrace Logger software (Unisense, Aarhus, Denmark). Evaluation of the recordings was performed offline: by applying linear regression analysis, the signals from the calibration solutions were fitted with a curve and the data were converted to pH values using linear interpolation [15,18,19]. As the technique allows stable continuous $pH_{brain}$ measurements reliably only for 3–4 hours, in different animals, different time windows were chosen to be assessed (baseline, PA and the first 4 hours of reventilation (n = 6), 8th-14th hours (n = 8) and 20th-24th hours (n = 3) of reventilation as presented in the Results).

## Induction of asphyxia

After surgery, a 1h recovery period allowed stabilization of the monitored physiological parameters prior to obtaining their baseline values. PA was induced by switching ventilation from medical air to a hypoxic-hypercapnic gas mixture (6% $O_2$, 20% $CO_2$, balance $N_2$) for 20 minutes, simultaneously reducing the RR to 15 1/min and stopping the fluid/glucose administration. Piglets (n = 13) were reventilated (RR: 30 1/min) with medical air for the remaining time of the experiment.

In three additional animals, before inducing PA, the effect of graded normoxic hypercapnia on $pH_{brain}$ was evaluated by 5% step-wise increases in inhaled $CO_2$ from 0% to 20%, for 7 to 8 min each. After the graded hypercapnia, normocapnia was restored for 30 min. These animals were euthanized with an overdose of pentobarbital sodium (300 mg, Release; Wirtschaftsgenossenschaft deutscher Tierärzte eG, Garbsen, Germany) at 2 hours after PA.

## Neuropathology

The objective of the neuropathology examination was to test if the asphyxia-induced neuronal injury was similar to what we reported previously using this PA/HIE model at 24 hours after asphyxia [11]. Accordingly, out of the 13 piglets exposed to PA, only those animals which were maintained for 24 hours (n = 8) were included. The brains of the eight anesthetized animals were perfused with cold (4°C) physiological saline through the catheterized common carotid arteries 24 hours after the end of asphyxia. The brains were gently removed and immersion-fixed in 4°C, 4% paraformaldehyde solution before further processing. Paraffin embedded, 4 µm sections were stained with haematoxylin-eosin, and neuropathology was assessed by light microscopic evaluation (Leica Microsystems, Wetzlar, Germany). Damaged neurons were identified using the major hallmarks of dark eosinophilic cytosol, as well as pyknotic or disrupted nuclei. The degree of cerebrocortical neuronal damage in the frontal, parietal, temporal, and occipital cortices was determined adapting a previously published scoring system [11,20]. Briefly, the pattern of neuronal injury (none < scattered < grouped < panlaminar) was determined in 20–20 non-overlapping fields of vision under 20x magnification in each assessed cortical region. Then, scores (0–9) were given to each region based on the frequency (% of 20 examined fields) of the most severe pattern of injury observed. The neuronal damage in the putamen, thalamus and the hippocampal CA1 regions was assessed with cell counting in non-overlapping areas (in 5-5-3 fields of vision respectively; under 200x magnification) as in [11,21]. Neuronal injury in these regions was expressed as the percentage of damaged neurons.

## Statistical analysis

Results were analysed offline and plotted using SigmaPlot (v12.0, Systat Software Inc., San Jose, CA, USA) or a MATLAB environment (Mathworks Inc., Natick, MA, USA). Neuropathology scores are expressed as median, 25–75 and 5–95 percentiles. All other data are expressed as mean±SEM. Normality was tested with the Shapiro-Wilk test. The correlation between $P_aCO_2$ and $pH_{brain}$ data were calculated with MATLAB's polynomial curve fitting. Parametric data were compared with one-way repeated measure of analysis of variance (RM ANOVA) followed by the Student-Newman-Keuls *post hoc* test. Level of significance (p) was set at 0.05.

## Results

### $pH_{brain}$ changes during PA and HIE development

Induction of PA elicited a reduction in $pH_{brain}$ that continued without levelling off over the 20 min insult, during which $pH_{brain}$ dropped from the baseline value of 7.21±0.03 to 5.94±0.11 by the end of asphyxia (n = 6; Fig 1A). Upon reventilation, $pH_{brain}$ was restored to 7.0 in 29.4±5.5 minutes with subsequent stabilization at a level that was virtually indistinguishable from the original baseline. Thereafter, from 2 h onwards $pH_{brain}$ remained slightly below baseline (on average, by 0.10±0.02 pH units) without showing any marked alterations at any observed time point within the 24-hour follow-up period (Fig 1B).

During the 20 min asphyxia period, $O_2$ saturation fell to below 30% and HR increased from about 140 1/min to nearly 200 1/min within the first 2 min, after which they showed no major changes, whereas the response in MABP was biphasic with a transient rise to about 90 mmHg followed by a slower decrease to below baseline (Fig 2). A rapid recovery of $O_2$ saturation to above 80% was seen during the first 2–3 min of reventilation, paralleled by transient increases in HR and MABP to above 240 1/min and 80 mmHg, respectively, after which the signals gradually recovered towards their normal values.

Longer-term monitoring of the physiological parameters reflected well the expected effects of PA and reventilation. Blood hemoglobin concentration (baseline: 8.0±0.3 g/dl; time course not illustrated), core body temperature (controlled by heating, see Methods), $O_2$ saturation, MABP and HR were within the normal range at baseline before PA and the values were not significantly different from baseline throughout the survival (Fig 3A–3D). In addition to the pulsoxymetry data, arterial blood gas analysis at the end of PA also confirmed the development of central haemoglobin desaturation (from 94±4 to 13±4%) along with severe acidosis, hypoxia and hypercapnia (Fig 3E–3H). Indeed, the fall in the arterial blood pH ($pH_a$) from 7.53±0.03 to 6.79±0.02 was substantial and paralleled by a rise in $P_aCO_2$ to 160±6 mmHg, however, $pH_a$ remained more than 0.8 pH unit higher than $pH_{brain}$. Blood glucose and lactate levels were also profoundly raised (Fig 3I and 3J) [22], indicating the metabolic response to PA. The large drop in base excess by 17.4±1.5 mmol/L and reduction in bicarbonate concentration (Fig 3K and 3L) together with the low $pH_a$ (6.79) at the end of PA indicate that asphyxia developed much beyond the key clinical criteria of severe BA in human neonates (pH <7.0 and base deficit ≥12 to 16 mmol/L [3,4]). Reventilation quickly restored normoxia in arterial blood, but $P_aCO_2$ levels remained slightly elevated, although the change was statistically significant only at 4 hours. Base excess was already normalized by this time point, and $pH_a$ returned to 7.39 ±0.02, normal for piglets [23]. In a similar fashion, blood glucose and lactate levels also returned to baseline by 4 hours, although both were still significantly elevated at 1 hour after PA.

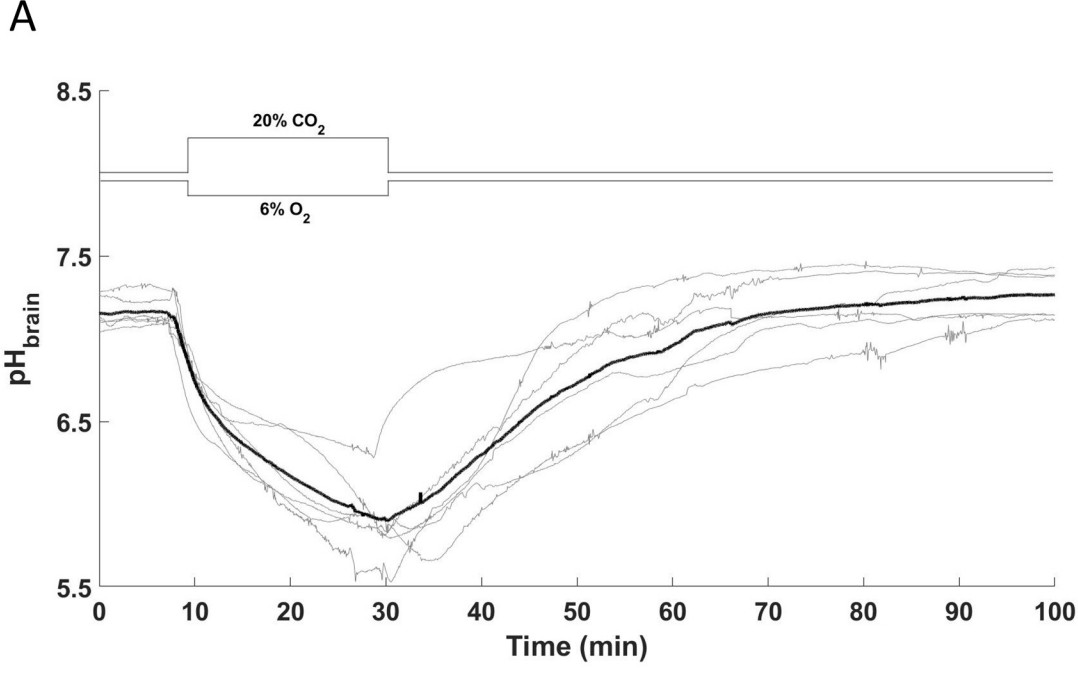

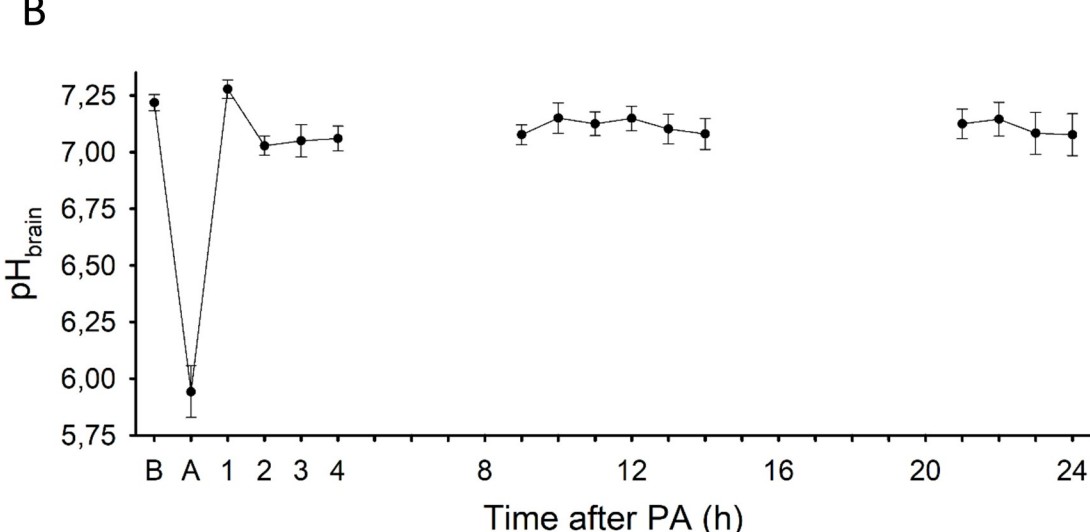

**Fig 1. pH$_{brain}$ changes during PA and the reventilation period.** (A) pH$_{brain}$ changes (grey lines–individual tracings, bold black line–mean) are plotted 10 min prior the onset asphyxia, during asphyxia, and the first hour of reventilation. pH$_{brain}$ fell progressively during asphyxia showing severe cerebrocortical acidosis, by the end of the insult the pH drop exceeded 1.0 pH unit virtually in all asphyxiated animals. Reventilation quickly restored pH$_{brain}$ to baseline levels. (B) pH$_{brain}$ alterations upon PA and over the 24-hour period after PA: after recovery from the PA-induced severe acidosis, no further significant pH$_{brain}$ alterations were detected at the selected time intervals (n = 6 at PA and between 1–4 hours; n = 8 between 8–14 hours and n = 3 between 20–24 hours; respectively). B: baseline, A: at the end of 20 minute PA. Panel B data points are presented as mean±SEM. *p<0.05 *vs*. baseline.

Neuropathology analysis confirmed the development of HIE by revealing medium/severe neuronal damage in all examined neocortical regions (Fig 4A and 4C), and also in the hippocampal CA1 region, the thalamus and the putamen (Fig 4B). These findings are in accordance with previously reported neuronal injury using the same PA stress [11].

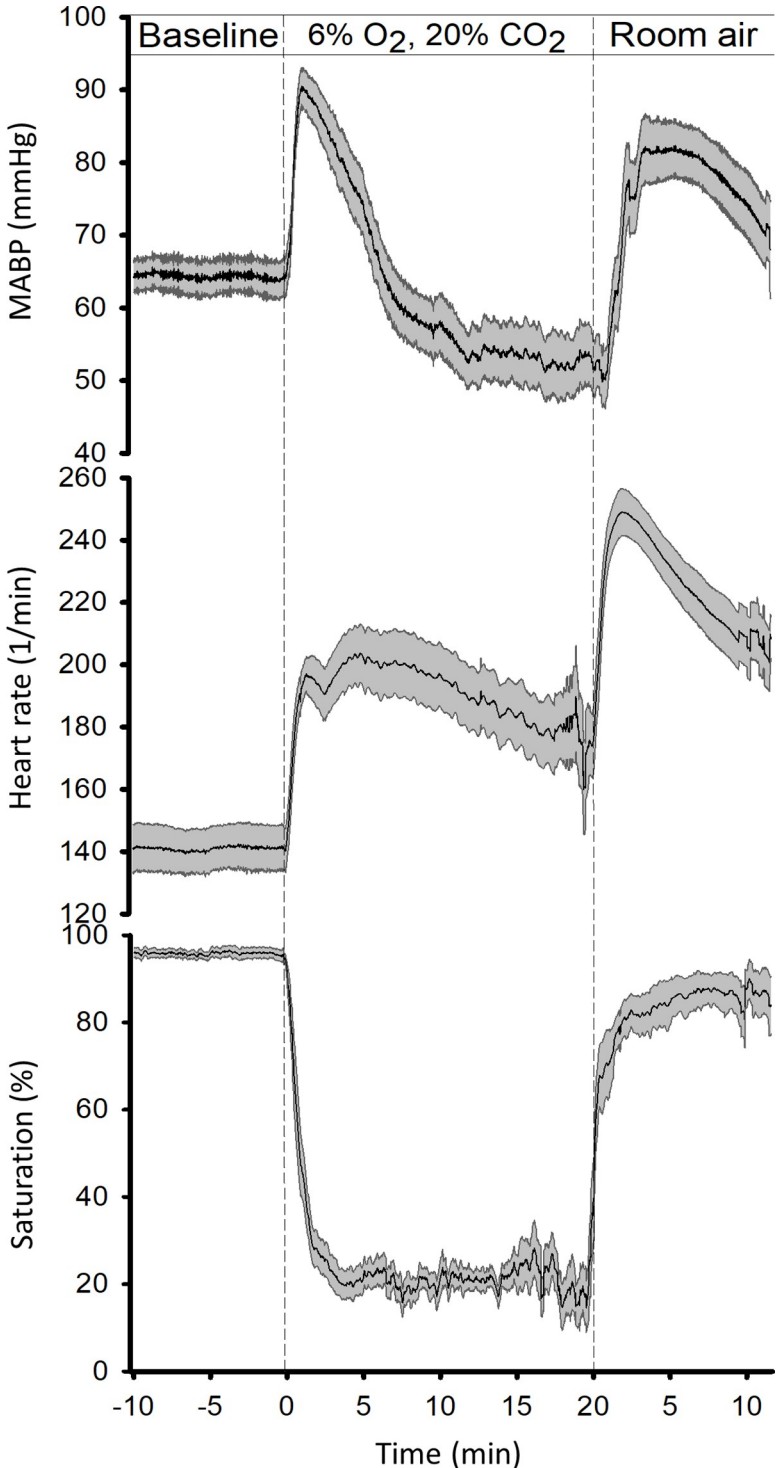

**Fig 2. Hemodynamic and oxygenation changes during PA.** In all panels, solid lines indicate the mean and the grey shaded area the SEM. (n = 13). After obtaining baselines, PA was initiated resulting first in the transient elevation of mean arterial blood pressure (MABP, top panel) and heart rate (HR, middle panel) accompanying the rapid fall in blood oxygen saturation (pulsoxymetry, bottom panel). While blood oxygen saturation remained ~20% throughout the asphyxia, after the initial peak, MABP fell continuously below baseline, whereas the drop in HR was more moderate and HR remained elevated. Reventilation with air quickly restored oxygen saturation and induced a second transient elevation in both MABP and HR.

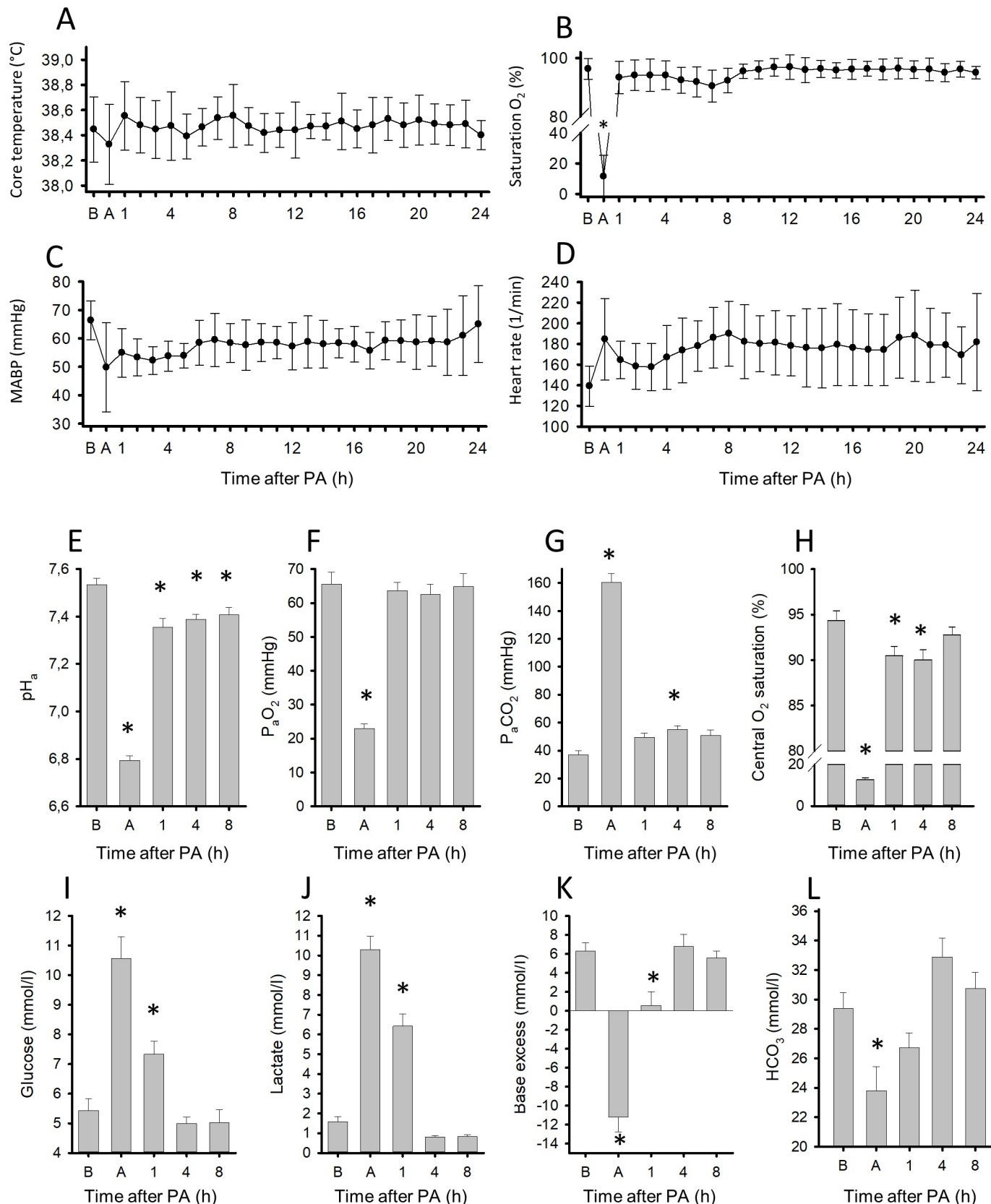

**Fig 3. Physiological parameters and blood chemistry changes during PA and the subsequent 24h reventilation period.** Core temperature of the animals (n = 13) was maintained in the physiologic range (38.5±0.5˚C) during the whole experiment (A). Blood oxygen saturation by pulsoxymetry (B) and mean arterial blood pressure (MABP, (C)) returned to baseline levels soon after asphyxia, however, the heart rate remained moderately elevated (D). Arterial blood gas analysis revealed that asphyxia resulted in severe acidosis (E), hypoxemia (F), hypercapnia (G), and central (arterial blood) desaturation (H). Plasma glucose (I) and lactate levels (J) were markedly elevated along with large drops in base excess (K) and significant reductions in blood bicarbonate concentrations (L). Reventilation restored most of the deranged parameters by 4 hours, and they were not significantly different from baseline levels afterwards, except for pH that was restored to the normal values [23] and not to the slightly alkalotic baseline. B: baseline, A: at the end of 20 minute PA. Bars and whiskers represent mean ±SEM, *p<0.05 *vs*. baseline values.

## pH$_{brain}$ changes during graded normoxic hypercapnia

These experiments were performed to evaluate the contribution of the respiratory component to the pH$_{brain}$ changes recorded during PA. Step-wise, 5% increases in inhaled $CO_2$ (Fig 5A) resulted in proportional step-wise reductions in pH$_{brain}$ (Fig 5B and 5C). Arterial blood gas

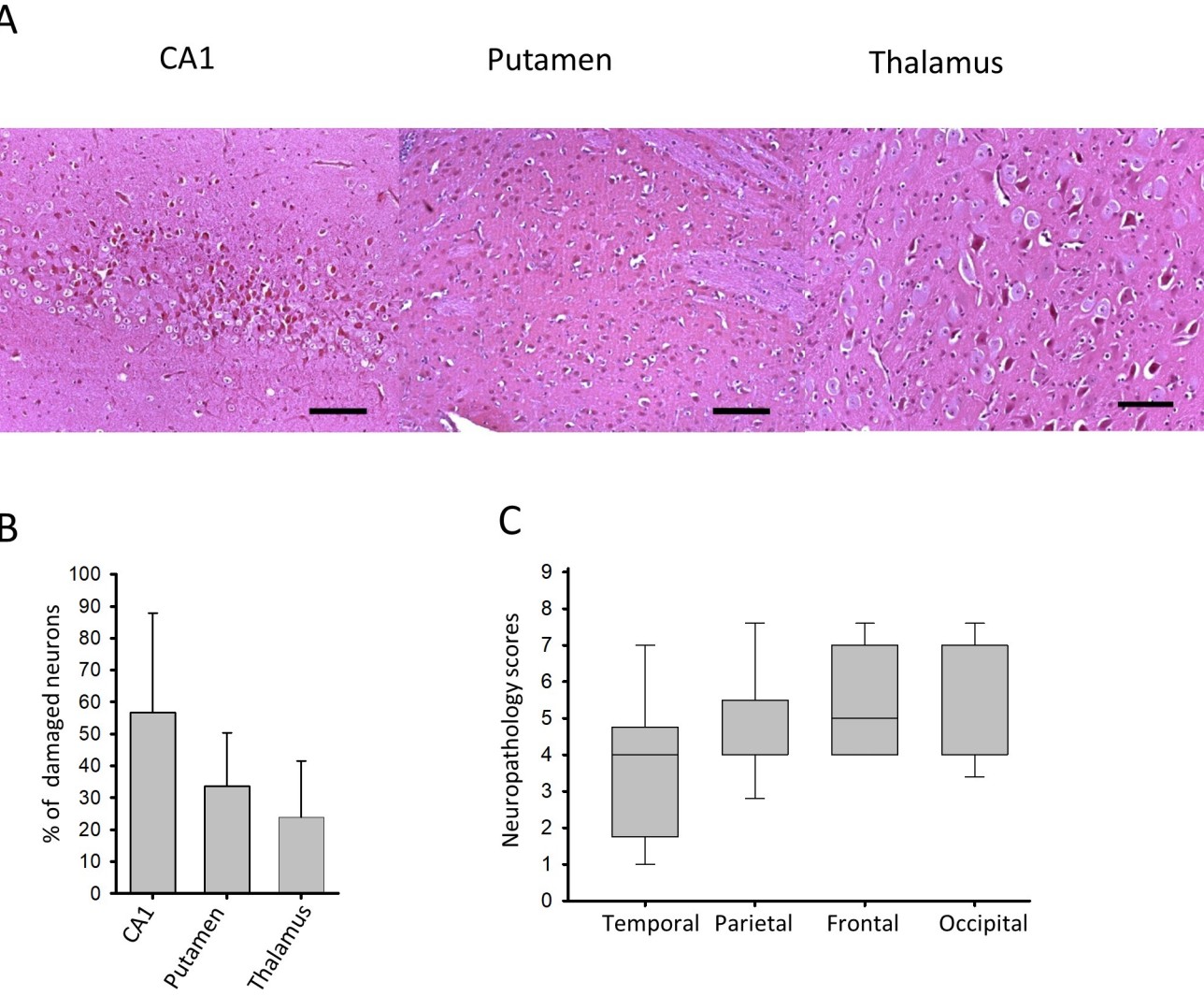

**Fig 4. Neuronal injury evaluated at 24 hours after PA.** (A) Representative photomicrographs showing injured red neurons at 24-hours of after asphyxia in the CA1 hippocampal region, the putamen, and the thalamus (scale bar: 100μm) (B) Cell counting revealed moderate neuronal damage in these regions (n = 8, mean±SEM). (C) Asphyxia also induced moderate/severe neocortical damage shown by medium/high neuropathology scores (lines, boxes, and whiskers represent the median, the 25th-75th, and the 5th-95th percentiles, respectively).

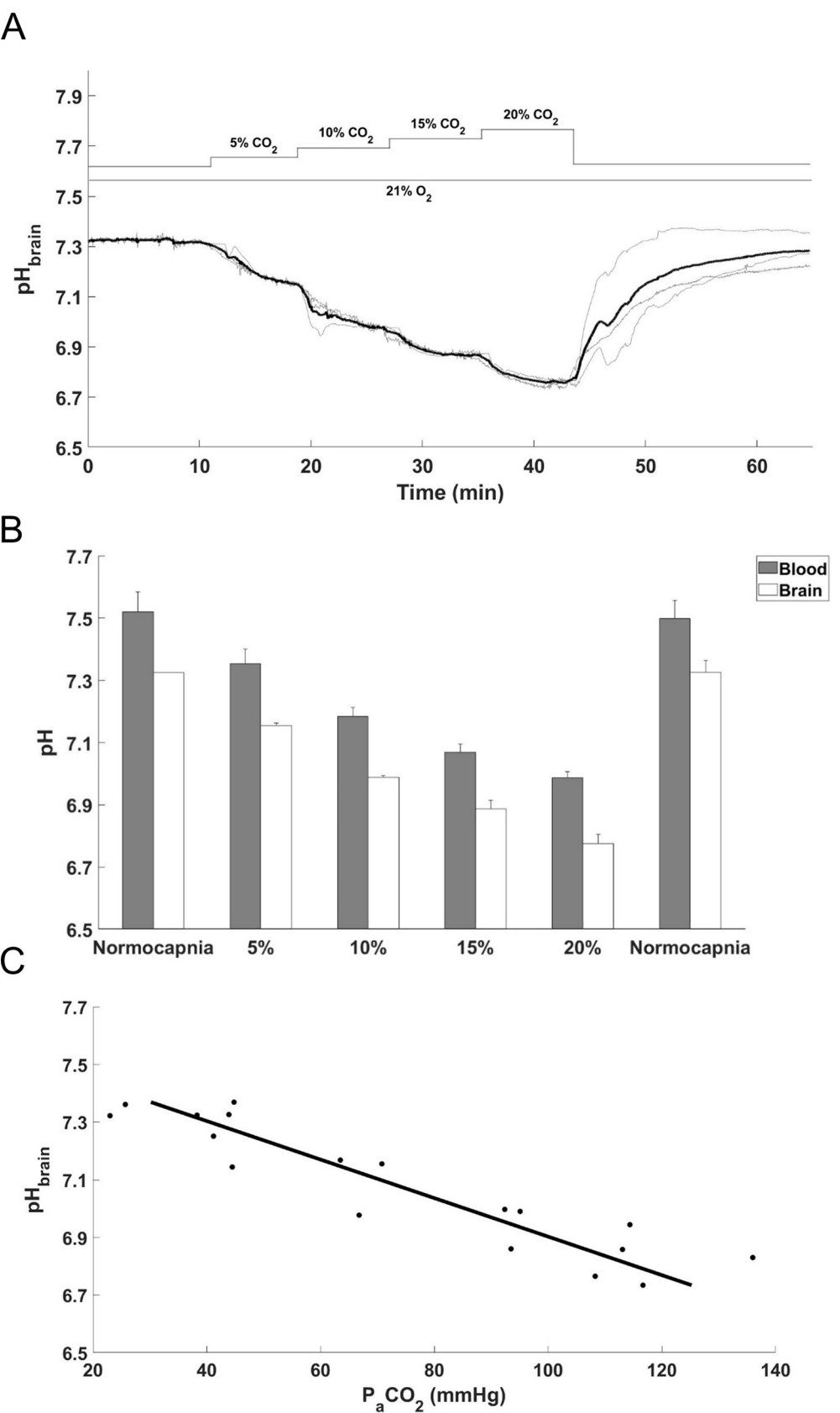

**Fig 5. Cerebral acidosis induced by graded normoxic hypercapnia.** (A) $pH_{brain}$ changes during graded normoxic hypercapnia induced by ventilation with 5–20% $CO_2$ followed by restoration of normocapnia (grey lines–individual tracings, bold black line–mean). (B) Simultaneous reductions in $pH_a$ and $pH_{brain}$ during graded normoxic hypercapnia were in accordance with the elevations in $P_aCO_2$, the values were (mmHg) 5%: 59.6±7.8, 10%: 84.7±9.0, 15%: 107.0 ±6.7, 20%: 120.3±8.2, (mean±SEM; n = 3). Corresponding $P_aO_2$ values were (mmHg) 5%: 75.5±12.8, 10%: 78.2±11.3, 15%: 85.5±8.2, 20%: 84.5±9.8, (mean±SEM; n = 3). (C) Linear regression shows close correlation between $P_aCO_2$ and $pH_{brain}$ ($R^2$ = 0.8546) values obtained during graded hypercapnia (5–20% $CO_2$).

analysis revealed similar graded reductions in $pH_a$, thus the difference between $pH_{brain}$ and $pH_a$ remained unchanged during hypercapnia (Fig 5B). Maximal changes were observed during the inhalation of 20% $CO_2$ when $pH_a$ dropped from 7.52±0.06 to 6.98±0.02, whereas $pH_{brain}$ from 7.32±0.01 to 6.77±0.02. The $pH_{brain}$ and $pH_a$ changes were fully reversed by restoration of normocapnia.

## Discussion

In this study we present *in vivo* real-time recorded $pH_{brain}$ values during PA and the subacute phase of HIE in a translational piglet PA/HIE model. The major findings of the present study are the following: (1) our experimental model elicited PA reflecting its major hallmarks and triggered neuronal injury corresponding to moderate/severe HIE revealed by neuropathology; (2) cerebrocortical $pH_{brain}$ dropped drastically in response to PA, the acidosis exceeding one pH unit in the brain compared to the arterial blood; (3) reventilation/reoxygenation allowed the restoration of $pH_{brain}$ to baseline levels, then $pH_{brain}$ remained stable around baseline levels without major shifts over the 24h observation period.

Every PA/HIE model using postnatal animals inevitably carries the limitation that PA is induced after and not before/during the cardiorespiratory adaptation to extrauterine life. However, we believe that in our study this disadvantage has been minimized by two factors. First, we used truly newborn piglets, thus their vulnerability to PA was least likely to have considerably changed by postnatal development. Second, we elicited PA that resulted in severe enough arterial pH and blood gas alterations that are known to signal human HIE development. This statement is justified by a clinical study that reported significant differences in the degree of acidosis ($pH_a$: 6.75±0.18 vs 6.90±0.18) and hypercapnia ($P_aCO_2$: 141±37 vs $P_aCO_2$: 94±22 mmHg) between asphyxiated babies presenting vs not presenting with HIE [24]. In our model the corresponding values were $pH_a$: 6.79±0.02 and $P_aCO_2$: 160±6 mmHg, closely matching the data from human HIE patient group. Interestingly, virtually identical values from umbilical artery blood samples ($pH_a$: 6.69±0.04 and $P_aCO_2$: 156±4 mmHg) were obtained in piglets undergoing spontaneous PA during delivery also indicating the translational value of the piglet as a PA/HIE model species [25]. We have developed and used this PA/HIE protocol to test the putative neuroprotective effect of molecular hydrogen [11]. We report virtually identical neocortical, hippocampal, and subcortical neuronal injury in the present study compared to those in [11]. Based on the similar neuropathology findings, we can assume that the surgical manipulations associated with $pH_{brain}$ measurements did not affect HIE development in this study, lending support to the translational value of the present $pH_{brain}$ findings.

The importance of $pH_{brain}$ determining neurological outcome following hypoxic-ischemic stress has long been acknowledged [26]. However, quantitative data on $pH_{brain}$ changes during/after PA in piglet PA/HIE models are very scarce in the literature [16,27,28]. Bender *et al.* [16] assessed $pH_{brain}$ using a similar technique in 1–3 days old piglets. PA was induced also with a hypoxic-hypercapnic gas mixture (5–8%$O_2$-7%$CO_2$) for 30 min. At the end of PA, the measured $pH_{brain}$ was 6.26±0.14, which is about 0.3 pH unit higher than in our study and likely

at least part due to the much lower $P_aCO_2$ values (61±1 *vs.* 160±6 mmHg [16] *vs* present study, respectively). After PA, the reventilation commenced with 100% $O_2$ ventilation, in addition, sodium bicarbonate (2 mEq/kg, iv) was infused for rapid correction of arterial pH. Both of these interventions are likely to decrease the translational value of the study by Bender *et al.* [16] as they are not included in current guidelines of neonatal care, and they may have affected the recovery of $pH_{brain}$ that was completed in 90 min. Moreover, there was no significant neuronal injury compared to sham operated animals, except experiments where PA was combined with hemorrhagic hypotension, suggesting that the applied PA as such was not severe enough to elicit HIE. Corbett *et al.* [27] used MRS in 8±3 day old piglets to determine $pH_i$ during and after ischemia but not genuine PA. Incomplete cerebral ischemia was elicited by combining bilateral carotid artery occlusion and hemorrhagic hypotension for 25 min followed by 90 min of reperfusion [28]. Severe acidosis developed during the ischemia that was dependent on blood glucose levels during the stress. Brain $pH_i$ dropped below 5.6 in fed piglets that responded to ischemia with hyperglycemia (9.4-15 mmol/L), whereas in fasted piglets responding with hypoglycaemia (1.4-2.6 mmol/L) the nadir at the end of ischemia was mere $pH_i \cong 6.6$. Upon reperfusion, $pH_i$ was again normalized within the 90 min observation period.

From the above discussed studies it is clear that higher levels of hypercapnia, cerebral blood flow, and blood glucose all promote the development of cerebral acidosis during PA. Increases in $P_aCO_2$ to 140–160 mmHg observed both in human and piglet PA will alone reduce $pH_i$ to 6.5–6.6 under normoxic conditions [29]. Also in the present study, we provide evidence that under normoxic conditions the inhalation of 20% $CO_2$ alone ($P_aCO_2$:120 mmHg) results in a $pH_{brain}$ drop to 6.8. Using linear regression (Fig 4C), we then calculated that the developing hypercapnia (at $P_aCO_2$: 160 mmHg) alone results in a $pH_{brain}$ drop to 6.50 in our PA model, in full agreement with previous results. Further acidification during PA will be dominantly determined by the increases in the rate of anaerobic glycolysis fuelling the subsequent lactic acid production. The rate of glycolysis is limited by glucose delivery to the hypoxic brain, as its reduction by either reducing cerebral blood flow [16] or blood glucose levels [28] attenuated the development of acidosis. We have previously shown that in our PA model significant cerebral ischemia does not develop [11] as in most animals the drop in MABP during the 20 min PA (Fig 4) does not reach the lower limit of blood flow autoregulation [30]. Admittedly, PA in our model has not been long enough for the development of hypotension and bradycardia commonly seen in severely asphyxiated infants, however, it can nicely represent the PA phase before the cardiovascular adaptation mechanisms are exhausted and likely the most pronounced $pH_{brain}$ alterations occur. In summary, our current study provides new compelling experimental evidence that not only cerebral ischemia but *bona fide* PA combining clinically relevant levels of hypoxia, hypercapnia and hyperglycemia is sufficient to elicit cerebral acidosis that is severe enough ($pH_{brain} < 6.0$) to strongly affect neuronal viability [31]. The developing $\cong 0.8$ pH unit difference between $pH_{brain}$ and $pH_a$ signals a >6-fold $H^+$ gradient across the blood-brain barrier that is in compliance with previous findings that the blood-brain barrier is mature in newborn pigs and not severely compromised by PA [32].

Alterations in $pH_{brain}$ may play important pathophysiological roles not only during acute PA, but also after reventilation/reoxygenation. Previously, $pH_{brain}$ was reported to be stable in piglets after restoration from PA, but it was not followed up beyond four hours of recovery [16]. In contrast, our current study extended the post-asphyxia observation period to 24 hours. We found no major secondary alterations in $pH_{brain}$ in this piglet PA/HIE model during the 24-hour period after PA. Our present findings are in compliance with previous $pH_i$ data obtained with MRS in a piglet HIE model using hypoxic-ischemic stress instead of PA [33]. In this study, $pH_i$ remained at baseline levels after restoration from the ischemic stress for 48 hours, furthermore, the development of secondary energy failure was not reflected in $pH_i$

alterations in this time period. In a human MRS study, $pH_i$ was also reported to be normal (7.13±0.05) in asphyxiated normocapnic newborns during the first day of life [34], in accordance with our present study. Importantly, brain alkalosis developed during the following days, and then persisted for weeks, even months [13,34]. However, we would like to point out that our PA/HIE piglet model represents the subgroup of asphyxiated babies that require intubation and respiratory support, but not those who breathe spontaneously. While the mechanically ventilated babies stay normocapnic, the spontaneously breathing babies often hyperventilate and may develop hypocapnia [24]. This response can reflect a relative hyperventilation, secondary to the induction of hypoxic hypometabolism resulting in reduced $CO_2$ production [35]. Reduction in $P_aCO_2$ due to hyperventilation tends to elevate $pH_{brain}$ and this notion is of special interest as hypocapnia has been identified as an independent risk factor for adverse neurological outcome [36,37]. In piglets, moderate hypocapnia was sufficient to elicit a reduction in cerebral perfusion with a simultaneous increase in pH and lactate levels even in the absence of asphyxia [38]. Indeed, our findings indirectly suggest that maintenance of normocapnia/ slight hypercapnia may prevent secondary $pH_{brain}$ alterations. In a recent work aimed to establish a translationally valid small-animal model of birth asphyxia in rats and guinea pigs, higher $P_aCO_2$ levels during simulated birth asphyxia and gradual restoration of normocapnia afterwards were found to elicit beneficial effects on cerebral metabolic acidosis, oxygen and lactate levels [39]. Clearly, further studies are warranted to explore these effects in the piglet model. Our current study had some additional limitations. We collected $pH_{brain}$ data from the neocortex using only one level of asphyxia, leaving other brain regions and levels of stress unexplored.

## Conclusions

Our translational piglet PA/HIE model reproduces all major hallmarks of birth asphyxia and elicits significant neuronal injury without employing carotid artery occlusions and/or hemorrhagic hypotension. In this model, $pH_{brain}$ drops below 6.0, $\cong$0.8 pH unit lower than $pH_a$ during PA, establishing a pathogenetic role of severe acidosis in neuronal injury. However, secondary $pH_{brain}$ alterations after restoration of baseline levels were not observed during the 24-hour subacute period, perhaps due to the prevention of secondary hypocapnia by controlled mechanical ventilation.

## Author Contributions

**Conceptualization:** János Németh, Viktória Varga, Viktória Kovács, Valéria Tóth-Szűki, Kai Kaila, Juha Voipio, Ferenc Domoki.

**Formal analysis:** Gábor Remzső, János Németh, Kai Kaila, Juha Voipio, Ferenc Domoki.

**Methodology:** Gábor Remzső, János Németh, Viktória Varga, Viktória Kovács, Valéria Tóth-Szűki, Kai Kaila, Juha Voipio, Ferenc Domoki.

**Supervision:** János Németh, Viktória Varga, Viktória Kovács, Valéria Tóth-Szűki, Kai Kaila, Juha Voipio, Ferenc Domoki.

**Validation:** Kai Kaila, Juha Voipio, Ferenc Domoki.

**Visualization:** Gábor Remzső.

**Writing – original draft:** Gábor Remzső.

**Writing – review & editing:** Kai Kaila, Juha Voipio, Ferenc Domoki.

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
