## [Decision Letter · Decision Letter 0]

15 Apr 2020

PONE-D-20-01890

Brain interstitial pH changes in the subacute phase of hypoxic-ischemic encephalopathy in newborn pigs

PLOS ONE

Dear Mr. Remzső,

Thank you for submitting your manuscript to PLOS ONE. After careful consideration, we feel that it has merit but does not fully meet PLOS ONE’s publication criteria as it currently stands. Therefore, we invite you to submit a revised version of the manuscript that addresses all the points raised during the review process.

Both reviewers found the study interesting and indicated only minor issues to be corrected. Among others both of them made remarks that the neuropathological changes need to be described in more details.

We would appreciate receiving your revised manuscript by May 30 2020 11:59PM. To enhance the reproducibility of your results, we recommend that if applicable you deposit your laboratory protocols in protocols.io, where a protocol can be assigned its own identifier (DOI) such that it can be cited independently in the future. For instructions see: http://journals.plos.org/plosone/s/submission-guidelines#loc-laboratory-protocols

We look forward to receiving your revised manuscript.

Kind regards,

Mária A. Deli, M.D., Ph.D.

Academic Editor

PLOS ONE

Journal Requirements:

2. Please include more information about the company who provided the experimental animals for the study, ie company address.

3. We noticed you have some minor occurrence(s) of overlapping text with the following previous publication(s), which needs to be addressed:

https://doi.org/10.1016/j.mvr.2017.05.006

In your revision ensure you cite all your sources (including your own works), and quote or rephrase any duplicated text outside the Methods section. Further consideration is dependent on these concerns being addressed.

Additional Editor Comments:

The review process for this particular study took longer than expected and that is usual due to difficulties to find reviewers. Based on my personal experience of handling over 300 papers, it belongs to a tiny minority of manuscripts. Altogether 17 experts were invited and 15 of them, including the reviewers suggested by the authors, rejected the invitations based on different reasons. I appreciate the patience of the authors.

Reviewers' comments:

Reviewer's Responses to Questions

**Comments to the Author**

1. Is the manuscript technically sound, and do the data support the conclusions?

Reviewer #1: Yes

Reviewer #2: Yes

2. Has the statistical analysis been performed appropriately and rigorously? 

Reviewer #1: Yes

Reviewer #2: Yes

3. Have the authors made all data underlying the findings in their manuscript fully available?

Reviewer #1: Yes

Reviewer #2: Yes

4. Is the manuscript presented in an intelligible fashion and written in standard English?

Reviewer #1: Yes

Reviewer #2: Yes

5. Review Comments to the Author

Reviewer #1: This is a well designed study that aimed to assess the alteration of brain pH under hypoxic-hypercapnic and normoxic-graded hypercapnic conditions in newborn pigs. The goal of the study was to work out an animal model for perinatal asphyxia.

The conduction and the methodology are logically designed and clearly, reproducibly described. In the results the authors described the time course of different parameters during and after hypoxic and normoxic hypercapnia and reventilation.

I have some comments and questions:

• In the methods section authors are mentioning that arterial blood samples were taken at different occasions during the study. Would it be possible to provide other systemic blood gas parameters beside pH? It would be intersting to see how for example systemic PaO2 and PaCO2 were changing during the procedure. Obviously they were registered because systemic PaCO2 is shown in figure 5C. Brain tissue pH may be dependent by systemic CO2 and O2 amounts beside metabolic change of the tissue itself.

• It is not clear for this reviewer whether neuropathological changes shown in figure 4 do represent which group of animals. According to the description, graded hypercapnia was induced in three animals and hypoxic hypercapnia in 13. The eight animals were from which group of animals? (probably out of the 13...?). Would this mean that neuropathological changes of the another 3 animals are not shon here? If this is the case were they different?

Thank you for having the opportunity to review this manuscript

Reviewer #2: General comment:

In the manuscript entitled ‘Brain interstitial pH changes in the subacute phase of hypoxic-ischemic encephalopathy in newborn pigs’, the authors reported their new method of well-characterized piglet perinatal asphyxia/neonatal hypoxic-ischemic encephalopathy (PA/HIE) model and provided the data of real-time brain interstitial pH changes. The manuscript is well written and provided fundamental and important data for PA/HIE. However, there are some points that the authors should address in the manuscript.

Please provide the discussion about the result of neuropathology (Figure 4). Any difference among the lesions (CA1, putamen, thalamus)?

The piglet PA/HAE model that authors described seems functional. The pHbrain drops during PA and causes neuronal injury, however, secondary pHbrain alterations were not observed. Authors described the reason of no secondary change is that controlled mechanical ventilation prevent from secondary hypocapnia. This is just speculation, but it is worth investigating further because it’s important for clinical application of therapy for HIE. Authors should refer to a bit more in discussion section.

Please also provide limitation section.

6. PLOS authors have the option to publish the peer review history of their article (what does this mean?). If published, this will include your full peer review and any attached files.

Reviewer #1: Yes: Béla Fülesdi

Reviewer #2: No

---

## [Author Response · Author response to Decision Letter 0]

2 May 2020

Response to the Academic Editor:

We thank the Editor for inviting us to respond to the critique by the two Reviewers. We are especially grateful for the effort that was apparently above average to procure the necessary reviews. Our point-by-point responses to the journal requirements / editorial comments are the following:

http://www.journals.plos.org/plosone/s/file?id=wjVg/PLOSOne_formatting_sample_main_body.pdf

and

http://www.journals.plos.org/plosone/s/file?id=ba62/PLOSOne_formatting_sample_title_authors_affiliations.pdf

Response: The figure legend titles and citation styles have been corrected as required. The Figure legends have been inserted after the paragraph in which they first appear as suggested (Fig 1: lines 194-203, Fig 2: lines 212-219, Fig 3: lines 239-250, Fig 4: lines 256-261, Fig 5: lines 270-277).

The figures have been uploaded to the Preflight Analysis and Conversion Engine (PACE) to ensure that figures meet PLOS requirements. We have attached all the PACE adjusted figures to the online platform.

The File names have been corrected.

Line 381: Conclusions section heading has been added.

Line 11-13: Author affiliations were corrected, short title has been removed.

2. Please include more information about the company who provided the experimental animals for the study, ie company address.

We included the company address in the manuscript:

Line 96: Pigmark Ltd., Co., H-6728, Rózsamajor út 13., Szeged, Hungary

3. We noticed you have some minor occurrence(s) of overlapping text with the following previous publication(s), which needs to be addressed:

https://doi.org/10.1016/j.mvr.2017.05.006

In your revision ensure you cite all your sources (including your own works), and quote or rephrase any duplicated text outside the Methods section. Further consideration is dependent on these concerns being addressed.

We have been baffled by this concern. We know the authors of the indicated study very well as we work for the same medical school. We could not find any overlap with the free on-line engines (like Dupli Checker) between the studies. That is not surprising since there is no overlap between the two studies concerning the hypothesis, the animal model species, the used techniques, or even the drugs, let alone the findings or the discussion. As both studies use in vivo animal experiments, and the groups share the same legal background concerning the regulations about it, we can only think there was an overlap between the description of this legal background and how the necessary permits in Hungary can be obtained. This section is in the Methods, which is according to journal policy such an overlap is tolerable, however, we still tried to rephrase somewhat this section to prevent even the hint of plagiarism. The possible overlapping text may be in lines 83-89, which describes these procedures. The rephrased sentences are the following: 

“…). Third, the National Food Chain Safety and Animal Health Directorate of Csongrád county, Hungary on behalf of the Hungarian Government issued the permit based on the ÁTET recommendation (permit nr: XIV./1414/2015). All animal experiments complied with (1) the guidelines of the Scientific Committee of Animal Experimentation of the Hungarian Academy of Sciences (updated Law and Regulations on Animal Protection: 40/2013. (II. 14.) Gov. of Hungary), (2) the EU Directive 2010/ 63/EU on animal protection used for scientific research, and (3) with the ARRIVE guidelines. .”

In case of any further such concern, please advise us immediately.

Response to Reviewer #1:

Reviewer #1: This is a well designed study that aimed to assess the alteration of brain pH under hypoxic-hypercapnic and normoxic-graded hypercapnic conditions in newborn pigs. The goal of the study was to work out an animal model for perinatal asphyxia.

The conduction and the methodology are logically designed and clearly, reproducibly described. In the results the authors described the time course of different parameters during and after hypoxic and normoxic hypercapnia and reventilation.

We thank the Reviewer for taking the time to evaluate our manuscript and for his/her helpful comments. We are also grateful for the appreciation of our work. 

I have some comments and questions:

#1. In the methods section authors are mentioning that arterial blood samples were taken at different occasions during the study. Would it be possible to provide other systemic blood gas parameters beside pH? It would be intersting to see how for example systemic PaO2 and PaCO2 were changing during the procedure. Obviously they were registered because systemic PaCO2 is shown in figure 5C. Brain tissue pH may be dependent by systemic CO2 and O2 amounts beside metabolic change of the tissue itself.

The physiological parameters of the animals exposed asphyxia were monitored, and the relevant data have been summarized in Figure 3. In Figure 3, 8 panels from Fig 3E to Fig 3L present the data obtained from the arterial blood samples, showing the full blood gas panel, and in addition, lactate and glucose data. The legend of Figure 5 reported the average PaCO2 levels, and in the revision we included the blood PaO2 data as well (lines 275-276).

#2. It is not clear for this reviewer whether neuropathological changes shown in figure 4 do represent which group of animals. According to the description, graded hypercapnia was induced in three animals and hypoxic hypercapnia in 13. The eight animals were from which group of animals? (probably out of the 13...?). Would this mean that neuropathological changes of the another 3 animals are not shon here? If this is the case were they different?

We are sorry for any discrepancies. Indeed, 13 piglets were exposed to asphyxia, and from those 8 animals were observed for 24 hours, and only those were processes for neuropathology examinations. The aim of the neuropathological examination was only to make sure that indeed asphyxia induced neuronal injury – HIE, to a similar extent that was reported in Ref 11. We made a number of changes in the manuscript to clarify this issue:

Line 156-159: “ The objective of the neuropathology examination was to test if the asphyxia induced neuronal injury was similar to what we reported previously using this PA/HIE model at 24 hours after asphyxia [11]. Accordingly, out of the 13 piglets exposed to PA, only those animals which were maintained for 24 hours (n=8) were included. “

Furthermore, we did not wish to perform neuropathology on the three piglets that were involved in the pHbrain measurements during graded normoxic hypercapnia. We do not expect that short-term normoxic hypercapnia that has been used even in human studies to test cerebrovascular reserve would cause measurable neuronal injury. Nevertheless, the short duration of these experiments would have excluded the possibility to detect neuronal injury with the technique employed. 

Response to Reviewer #2:

Reviewer #2: General comment:

In the manuscript entitled ‘Brain interstitial pH changes in the subacute phase of hypoxic-ischemic encephalopathy in newborn pigs’, the authors reported their new method of well-characterized piglet perinatal asphyxia/neonatal hypoxic-ischemic encephalopathy (PA/HIE) model and provided the data of real-time brain interstitial pH changes. The manuscript is well written and provided fundamental and important data for PA/HIE. However, there are some points that the authors should address in the manuscript.

We thank the Reviewer for his/her helpful comments and the appreciation of our work.

#1. Please provide the discussion about the result of neuropathology (Figure 4). Any difference among the lesions (CA1, putamen, thalamus)?

We modified the manuscript to clarify the role of the neuropathology. We included/modified statements in the Methods and the Results section, please see our response to Reviewer #1, concern#2. In addition, we inserted the following section in the discussion: 

Lines 301-306. “We have developed and used this PA/HIE protocol to test the putative neuroprotective effect of molecular hydrogen [11]. We report virtually identical neocortical, hippocampal, and subcortical neuronal injury in the present study compared to those in [11]. Based on the similar neuropathology findings, we can assume that the surgical manipulations associated with pHbrain measurements did not affect HIE development in this study, lending support to the translational value of the present pHbrain findings.”

#2. The piglet PA/HAE model that authors described seems functional. The pHbrain drops during PA and causes neuronal injury, however, secondary pHbrain alterations were not observed. Authors described the reason of no secondary change is that controlled mechanical ventilation prevent from secondary hypocapnia. This is just speculation, but it is worth investigating further because it’s important for clinical application of therapy for HIE. Authors should refer to a bit more in discussion section.

We agree with the Reviewer that this is a speculation at this point, and we thought it was worth a notion, therefore we did not wish to elaborate on it. However, we have added a few thoughts to this part of the Discussion and combined it with the requested limitation section above as well. The following section was inserted:

Lines: 366-379: “This response can reflect a relative hyperventilation, secondary to the induction of hypoxic hypometabolism resulting in reduced CO2 production [35]. Reduction in PaCO2 due to hyperventilation tends to elevate pHbrain and this notion is of special interest as hypocapnia has been identified as an independent risk factor for adverse neurological outcome [36,37]. In piglets, moderate hypocapnia was sufficient to elicit a reduction in cerebral perfusion with a simultaneous increase in pH and lactate levels even in the absence of asphyxia [38]. Indeed, our findings indirectly suggest that maintenance of normocapnia/ slight hypercapnia may prevent secondary pHbrain alterations. In a recent work aimed to establish a translationally valid small-animal model of birth asphyxia in rats and guinea pigs, higher PaCO2 levels during simulated birth asphyxia and gradual restoration of normocapnia afterwards were found to elicit beneficial effects on cerebral metabolic acidosis, oxygen and lactate levels [39]. Clearly, further studies are warranted to explore these effects in the piglet model. Our current study had some additional limitations. We collected pHbrain data from the neocortex using only one level of asphyxia, leaving other brain regions and levels of stress unexplored”

---

## [Decision Letter · Decision Letter 1]

14 May 2020

Brain interstitial pH changes in the subacute phase of hypoxic-ischemic encephalopathy in newborn pigs

PONE-D-20-01890R1

Dear Dr. Remzső,

We are pleased to inform you that your manuscript has been judged scientifically suitable for publication and will be formally accepted for publication once it complies with all outstanding technical requirements.

With kind regards,

Mária A. Deli, M.D., Ph.D.

Academic Editor

PLOS ONE

Additional Editor Comments (optional):

Reviewers' comments:

Reviewer's Responses to Questions

**Comments to the Author**

1. If the authors have adequately addressed your comments raised in a previous round of review and you feel that this manuscript is now acceptable for publication, you may indicate that here to bypass the “Comments to the Author” section, enter your conflict of interest statement in the “Confidential to Editor” section, and submit your "Accept" recommendation.

Reviewer #1: All comments have been addressed

Reviewer #2: All comments have been addressed

2. Is the manuscript technically sound, and do the data support the conclusions?

Reviewer #1: Yes

Reviewer #2: Yes

3. Has the statistical analysis been performed appropriately and rigorously? 

Reviewer #1: Yes

Reviewer #2: Yes

4. Have the authors made all data underlying the findings in their manuscript fully available?

Reviewer #1: Yes

Reviewer #2: Yes

5. Is the manuscript presented in an intelligible fashion and written in standard English?

Reviewer #1: Yes

Reviewer #2: Yes

6. Review Comments to the Author

Reviewer #1: The authors responded to all of my questions properly and they addressed all of my proposals. They indicated the changes in their response and in the manuscript.

Reviewer #2: Authors nicely revised their manuscript point by point according to reviewer’s comments. No additional comments.

7. PLOS authors have the option to publish the peer review history of their article (what does this mean?). If published, this will include your full peer review and any attached files.

Reviewer #1: No

Reviewer #2: Yes: Yoichi MOROFUJI

---

## [Editor Report · Acceptance letter]

20 May 2020

PONE-D-20-01890R1 

Brain interstitial pH changes in the subacute phase of hypoxic-ischemic encephalopathy in newborn pigs 

Dear Dr. Remzső:

I am pleased to inform you that your manuscript has been deemed suitable for publication in PLOS ONE. Congratulations! Your manuscript is now with our production department. 

With kind regards,

on behalf of

Dr. Mária A. Deli 

Academic Editor

PLOS ONE